# Determinants of Preventive Behaviors for COVID-19 in Japan

**DOI:** 10.3390/ijerph18199979

**Published:** 2021-09-23

**Authors:** Motoko Kosugi

**Affiliations:** College of Engineering, Academic Institute, Shizuoka University, Hamamatsu 432-8561, Japan; kosugi.motoko@shizuoka.ac.jp

**Keywords:** COVID-19, preventive behavior, risk perception, affect

## Abstract

As of June 2021, there have been more than 13,000 deaths in Japan due to the COVID-19 pandemic. Data from the Ministry of Health, Labor, and Welfare show that the mortality rate of COVID-19 greatly varies by age. In this study, using data from a questionnaire survey, an investigation was carried out to find differences in anxiety and risk perception, attitudes toward risk, and the frequency of implementation of countermeasures to infection among age groups that are prone to a greater risk of mortality, as well as the main factors that determine the frequency of implementation. Older people, who form a high-risk group, have a stronger tendency for anxiety and cautious attitudes toward COVID-19, and they more frequently implement preventive behaviors. The results of multiple regression analysis showed that the frequency of implementation of behaviors is determined not only by anxiety, cautious attitude, risk of aggravation to oneself, and perceived effectiveness of behaviors but also by regret, altruism, and conformity. In addition, almost no age-based gap was found between the determinants, suggesting that the motivation to take infection preventive behaviors is the same regardless of age.

## 1. Introduction

In Japan, the first state of emergency due to COVID-19 was declared on 7 April 2020, with several further declarations since. Under a state of emergency, severe restrictions are placed on the functioning of restaurants, participation in events, and use of public transportation facilities [1]. These restrictions have a serious impact on various aspects of people’s daily lives, including the avoidance of nonessential and nonurgent outings, the closure of schools and commercial facilities, and the reduction of medical services. However, unlike other countries, the Japanese government asks the public to wear masks and refrain from going out, but this is just a request, not obligatory, and there are no penalties.

Regarding the risk management of COVID-19, on 4 May 2020, the Ministry of Health, Labor, and Welfare (MHLW) announced practical examples of a “new lifestyle” that included four items, namely, (1) basic infection prevention measures for each person, (2) basic lifestyle for daily life, (3) lifestyle for each scene of daily life, and (4) new working style [2]. People’s behaviors are changing, with the use of masks in public remaining at a high level (62% in March 2020, 86% in May, and 89% in December) [3]. The Cabinet Secretariat website on “COVID-19 Countermeasures” shows that people’s movement has decreased in comparison with that prior to the pandemic, including the flow of people to train station areas and major tourist spots in each prefecture, the rate of use of stations, and the number of vehicles passing on highways [4].

Although many prior studies have focused on risk perception as one of the main factors determining people’s risk management behavior, it cannot be said that its effects are clear. Some findings suggest that there is a positive correlation between risk perception and behavior, but others show that greater risk perception does not necessarily lead to action even if the risk will presumably cause damage to one’s life and property. The former have shown that the perception of danger to oneself due to natural disasters and risk of illness promotes risk management behaviors [5,6,7,8,9,10]. However, others have reported that greater risk perception does not predict evacuation actions or behaviors such as the confirmation of evacuation routes [11,12,13,14,15].

Prior studies have shown that risk management behavior is influenced not only by risk perception and the perception of risk reduction effectiveness, which are thought to directly motivate the reduction of risk, but also by affect [16,17]. When people see others engaging in risk management behavior, they perceive it as a social norm whereby certain actions should be taken in certain situations, and when these actions are imitated, such risk management behavior is interpreted as arising from conformity. During the 2009 swine flu pandemic, the use of masks functioned as a social norm in Hong Kong [18], and it has been reported that the use of masks by Japanese people during the COVID-19 pandemic is a behavior that stems from conformity to social norms rather than the expectation of a risk reduction effect [19]. Further, it has been reported that behavioral choices are influenced by anxiety regarding risk and the anticipation of regret [10,20,21,22,23].

According to open data from the MHLW about the number of confirmed cases [24], by far, the most cases are among people in their 20s (141,292), followed by many people in their 30s (95,354), 40s (92,867), 50s (84,380), 60s (54,400), 70s (48,537), and 80s or above (48,497) (as of 12 May 2021). In contrast, the mortality rate is 0.0% among those aged 10–19, those in their 20s, and those in their 30s, whereas it is 0.1% for those in their 40s, 0.3% for those in their 50s, 1.3% for those in their 60s, 4.8% for those in their 70s, and 13.2% for those in their 80s and above. In other words, the mortality rate for COVID-19 greatly varies by age, with older people clearly being at greater risk, whereas younger people who have a much higher infection rate have a relatively low risk of mortality. Currently, there are many confirmed cases in the low-risk age group, whereas the number of confirmed cases among high-risk older people is relatively low.

This study identifies factors that contribute to the promotion of preventive behavior for COVID-19. Specifically, this study investigates whether people’s risk perceptions corresponded to actual risk assessments of age groups, and whether risk management behaviors, such as hand washing and wearing masks, depended on actual risk assessments. Moreover, it reveals not only risk perceptions and attitudes but also the role of emotional impact on the implementation of risk management behaviors.

## 2. Materials and Methods

Two cross-sectional online surveys were conducted with a nationally representative Japanese sample (*n* = 2400). The first survey was conducted on 2 and 3 October 2020 (low transmission period; 7-day average of 536 new infections for 30 September 2020, directly before the survey). The second survey was conducted on 12 and 13 January 2021 (high transmission period; 7-day average of 6446 new infections for 11 January 2021).

As regards the gender of respondents, for both the low and high transmission periods, there were 577 male respondents (48.1%) and 623 female respondents (51.9%). The average age was 52.6 years old (y/o) in the low transmission period and 51.8 y/o in the high transmission period.

A request for cooperation in the study was made by email to people aged 18 and above registered in the monitoring panel of a private research agency, and responses were received from 1200 people. Respondents were selected using quota sampling based on demographics by prefecture, gender, and age. Respondents to the first survey were not requested to cooperate in the second survey.

The questionnaire consisted of questions regarding COVID-19 risk characteristics (16 items, 7-point semantic differential scale), anxiety (15 items, 5-point scale), infection preventive behaviors being implemented (16 items, multiple selection), reason for implementation of these behaviors (13 items, multiple selection), attitudes (11 items, 5-point scale), and additional questions regarding vaccines in the second survey only. A descriptive analysis of the sample was carried out using percentages for the qualitative variables and means for the quantitative variables. Factor analysis (varimax rotation) was used for quantitative variables, namely, risk characteristics, anxiety, and attitudes. An analysis of variance was performed using the average of the factor scores for comparison between low and high transmission periods. Multiple regression analysis was performed to test the effects of each factor on preventive behaviors.

This survey was approved by the Ethics Review Committee of the university to which the author is affiliated.

## 3. Results

In this section, an analysis is made using the data from all 2400 respondents (1200 for each survey). In the following age-based analysis, the respondents were divided into three groups, namely, 18–39 y/o, 40–59 y/o, and 60 y/o and above. The number of respondents in each group was 342 in the 18–39 y/o group (28.5%), 385 in the 40–59 y/o group (32.1%), and 473 in the 60 y/o and above group (39.4%). The groups will be referred to in this text as the younger group, middle group, and older group, respectively.

The results below are organized first into factor analyses across different dimensions, followed by ratios of frequency of use of preventive measures, and finally in a regression analysis of identified factors against the frequency of use of preventive measures.

### 3.1. Perception of Characteristics of COVID-19

With regard to how the characteristics of COVID-19 risk are perceived, a factor analysis (varimax rotation) was conducted using the responses to 14 questions evaluated using the 7-point semantic differential method, and 2 factors were extracted (Table 1). The factor structure was the same regardless of the survey period. Five items had greater factor loading in the first factor, which was named “critical,” and two items had greater factor loading in the second factor, which was named “trivial.”

When investigating the differences among the age groups for each survey period with the factor scoring of each factor as the dependent variable, in both cases, a significant difference was seen in the “critical” factor (low transmission period: F (2, 1197) = 23.5, *p* < 0.001; high transmission period: F (2, 1197) = 26.8, *p* < 0.001), showing that higher age groups have a stronger tendency to recognize the seriousness of COVID-19. There was no gap between the age groups for the “trivial” factor.

### 3.2. Anxiety about COVID-19

Regarding anxiety about COVID-19, the respondents were asked to select the applicable number on a 5-point scale regarding 15 descriptors. When conducting a factor analysis for each survey period, two factors were extracted (Table 2). The two factors had the same structure, with the first factor comprising five items. These items are considered to be directly related to anxiety about COVID-19, so the factor was named “direct anxiety.” The second factor comprises six items that can be interpreted as anxiety about the impact arising from measures taken to prevent the spread of infection, leading to the factor being named “indirect anxiety.”

When investigating the differences among the age groups for each survey period with the factor scoring of each factor as the dependent variable, in both periods, a significant difference was seen in the “direct anxiety” factor (low transmission period: F (2, 1197) = 22.0, *p* < 0.001; high transmission period: F (2, 1197) = 26.4, *p* < 0.001), showing that higher age groups have a stronger tendency to have a direct sense of anxiety about COVID-19. There was no gap between the age groups for the “indirect anxiety” factor.

### 3.3. Attitudes toward COVID-19

Regarding attitudes toward COVID-19, the respondents were asked to select the applicable number on a 5-point scale regarding 11 descriptors. Based on a factor analysis, the first factor extracted comprised five items, and the second factor comprised four items (Table 3). Due to the content of the items, the first factor was named the “cautious” factor, and the second factor was named the “skeptical” factor.

In both survey periods, a significant age-based difference was seen in the attitude factors. Regarding the cautious factor (low transmission period: F (2, 1197) = 22.0, *p* < 0.001; high transmission period: F (2, 1197) = 24.8, *p* < 0.001), there was a stronger attitude of caution, with a greater perception of the risks of COVID-19, as the groups went up in age. Conversely, regarding the skeptical factor (low transmission period: F (2, 1197) = 16.5, *p* < 0.001; high transmission period: F (2, 1197) = 19.0, *p* < 0.001), there was a stronger attitude of skepticism about COVID-19, as the groups decreased in age.

### 3.4. Implementation of Preventive Behavior

In both the low and high transmission periods, there were more implementation items and a higher rate of implementation of preventive behaviors among the older group (Table 4). The number of countermeasures being implemented (out of a maximum of 14) during the low transmission period was 4.9 items for the younger group, 5.5 for the middle group, and 6.8 for the older group. During the high transmission period, it was 5.4 items for the younger group, 6.5 for the middle group, and 7.6 for the older group, indicating that more countermeasures are implemented in each age group during high transmission periods than low transmission periods and that significantly more countermeasures are taken as age increases in both survey periods (low transmission period: F (2, 1197) = 36.2, *p* < 0.001; high transmission period: F (2, 1197) = 38.1, *p* < 0.001). Confirmation of significant difference between all groups was based on Tukey’s HSD (honestly significant different) test.

The reasons for the behavior could be selected from among “perception of effectiveness,” “risk perception,” “regret,” “conformity,” and “altruism,” and multiple selections were allowed. As a result, looking at the respondents as a whole, more than 20% selected “Because I think it is effective in preventing infection (perception of effectiveness)” (low transmission period: 79.8%; high transmission period: 81.5%), “Because I don’t want to infect other people (altruism)” (low transmission period: 72.1%; high transmission period: 71.2%), “I don’t know if it has any effect, but I don’t want to regret not doing it (regret)” (low transmission period: 40.2%; high transmission period: 40.4%), “Because I am at high risk of aggravation (risk perception)” (low transmission period: 28.4%; high transmission period: 32.4%), and “Because everyone around me is doing it (conformity)” (low transmission period: 24.7%; high transmission period: 25.2%). The ratio of the selection of the reasons did not change depending on the infection status (ns in χ^2^ test for all).

Approximately 45% of the older group selected “Because I am at high risk of aggravation” in both the low transmission period and the high transmission period, whereas less than 20% of the younger group made the same choice. Approximately 30% of the young people selected “Because everyone around me is doing it,” whereas this option was selected by roughly 23% of the older and middle groups.

### 3.5. Determinants of the Implementation of Preventive Behaviors

In order to clarify the main factors that have an impact on preventive behaviors, a multiple regression analysis was performed with frequency of implementation as the dependent variable (Table 5). The number of implemented behaviors increased in line with an increase in anxiety about COVID-19 (direct and indirect), increase in cautious attitudes, decrease in skeptical attitudes, and increase in age. Further, the frequency of implementation was higher among women than men and higher in high transmission periods than in low transmission periods. Regarding the perception of risk characteristics, although it was found that the frequency of implementation decreases in line with a more trivial perception of the risk, there was no significant effect in the “critical” factor. The perception of risk caused by the difficulty of dealing with the situation, including the scientific uncertainties, the numerous simultaneous fatalities, and the impact on future generations, did not contribute to the frequency of implementation. Significant effects were caused by all of the items included in the model as the reasons for the implementation of behaviors, namely, “perception of effectiveness,” “altruism,” “regret,” “risk perception,” and “conformity,” and the implementation of preventive behaviors increased in line with the increase in the selection of each of these behavioral reasons.

When performing the same analysis by age group (Table 6), the model was more applicable to the younger group (adjusted R^2^ = 0.45) and middle group (0.44) than the older group (0.36). For all the age groups, a significant effect on the frequency of implementation of behaviors was found due to attitudes regarding COVID-19 (the cautious factor and skeptical factor), the perception of the effectiveness of behaviors, altruism, risk perception, regret, survey period, and gender. In the younger group, in addition to the main factors, there was a significant effect when anxiety about COVID-19 (direct anxiety) and conformity were selected as the reasons for the behavior. Similarly, in the middle group, there was a significant effect from the trivial perception of risk (negative effect) and direct anxiety, and in the older group, there was a significant effect from anxiety (indirect anxiety) regarding lifestyle restrictions due to the pandemic and conformity.

There were high parameter estimates in the model for the younger group for the cautious factor and regret, and the implementation of preventive behaviors increased in line with an increase in more cautious attitudes toward COVID-19 and a greater desire not to feel regret. There were high parameter estimates in the model for the middle group for direct anxiety about COVID-19 and regret, and the implementation of preventive behaviors increased in line with an increase in anxiety regarding infection and the strain on medical care and a greater desire not to feel regret. There were high parameter estimates in the model for the older group for skeptical attitudes, the cautious factor, and the perception of effectiveness, and the implementation of preventive behaviors increased in line with the increase in cautious attitudes and a stronger belief that these behaviors would be effective in preventing infection, whereas the frequency of implementation decreased in line with more skeptical attitudes. Although the above results are not significantly different from the model that explains the implementation of behaviors by age, there are age-based differences in the main factors that have a relatively large impact on the frequency of implementation.

### 3.6. Intention to Receive a Vaccine

In the high transmission period only (January 2021), questions were asked about the respondents’ intention to be vaccinated, which is thought to be the most effective preventive behavior (vaccination had not yet started in Japan at the time of responding to the questionnaire). In all age groups, most of the respondents chose “I will wait and see how the situation develops before I consider the vaccine” (younger group: 65.2%; middle group: 69.1%; and older group: 57.7%). However, a higher percentage of the older group selected “I will get vaccinated as soon as possible” than any of the other age groups (younger group: 19.1%; middle group: 15.8%; and older group: 33.4%). The response “I will not get vaccinated” was selected by 11.7% of the younger group, 10.7% of the middle group, and 7.2% of the older group, which was the lowest ratio.

Regarding the reason for vaccination behavior (multiple choice), in all age groups, the most common response among those who answered “I will not get vaccinated” and “I will wait and see” was “Because of fear of side effects (adverse reactions).” Among those in the older group who selected “I will wait and see,” in particular, fear of side effects was given as a reason by 81.2% of the respondents.

As regards the respondents who selected “I will get vaccinated as soon as possible,” in the younger group, more of them selected “Because I do not want to keep living in fear of the risk of infection” (67.7%) than “Because I and the family members I live with are at high risk of infection and aggravation” (38.5%). The same trend can be seen among other age groups, but the difference tends to diminish as age increases (middle group: 49.2%/67.2%; older group: 57.6%/71.5%).

“I don’t think vaccines are effective” and “I have doubts about all vaccines, not just those for COVID-19” were most common among respondents who selected “I will not get vaccinated” in all age groups. However, except for 47.1% of the older group (the highest ratio) who selected “I will not get vaccinated” and “I have doubts about all vaccines,” the rates of selection of the younger and the middle group were in the scope of 20–30%

## 4. Discussion

In this study, two online questionnaire surveys were conducted among men and women aged 18 and above throughout Japan, once in a low transmission period when the frequency of new COVID-19 confirmed cases was low and once in a high transmission period when the figures were increasing. As the risk of mortality of COVID-19 greatly differs by age, in this study, the respondents were divided into three age groups, and an investigation was conducted into the differences in their perceptions and attitudes toward COVID-19, their sense of anxiety, implementation of preventive behaviors, and so forth. As a result, it was found that as age increases, so do risk perception and sense of anxiety, cautious attitudes about COVID-19, and preventive behaviors, in proportion to the greater risk of mortality. In all age groups, there was an increase in the frequency of implementation of preventive behaviors during the high transmission period when confirmed cases were increasing in comparison with the low transmission period, which suggests that behavior changes according to the environment regardless of age. Further, regarding vaccination, a greater proportion of the older group than any other group responded with “I want to get vaccinated as soon as possible.” These results suggest that, in general, preventive behaviors are determined by a greater risk of infection and infection status in the community.

Based on a multiple regression analysis that clarified the main factors determining the number of preventive behaviors, regardless of infection status in the community, it was found that the frequency of implementation increases in line with more cautious attitudes, with COVID-19 being perceived as a serious problem in society, and that the frequency of implementation decreases in line with more skeptical attitudes, in terms of not wanting to change one’s behavior to deal with the risk and perceiving that the risk is being exaggerated. As there was a significant effect from personally being at high risk of aggravation, it was found that the perception of the danger of being personally affected leads to action. Not only risk perception but also the effectiveness of behaviors and variables related to affect factors, such as regret, altruism, and conformity, caused a significant increase in implemented behaviors. As stated in the background of the study, many prior studies have shown that affect influences decision making and behavior, which is consistent with the results of this study. Although the frequency of implementation varies by age, most of the main factors that determine the frequency of implementation are the same for all groups.

The finding that high-risk perception contributes to the implementation of preventive behaviors is consistent with the findings of previous studies that perception of high risk of infection has positive effects on behaviors such as hand washing and social distancing [25,26] and that adults with chronic diseases are more likely than adults without chronic diseases to adhere to preventive behaviors such as mask wearing and hand washing [27]. The finding that the implementation of preventive behaviors is related to risk perception indicates that there is room for improvement in risk communication implemented by governments and experts. However, as mentioned in Section 1, some studies support the view that risk perception does not always promote risk management behavior. Scholz and Freund [28] show that not risk perception but rather self-efficacy and response efficacy promote preventive behaviors. A study by Lin et al. [29] also shows that self-efficacy contributes to preventive behavior. Therefore, we should pay attention not only to the risk perception and emotional factors examined in this study but also to the efficacy of our involvement in risk management.

In addition, in terms of the characteristics of each age group, in the younger group, cautious attitudes and regret made a major contribution to the frequency of implementation of behaviors; in the middle group, a major contribution was made by direct anxiety and regret; and in the older group, a major contribution was made by cautious attitudes, skeptical attitudes (negative effect), and the perceived effectiveness of preventive behaviors. Based on these results, it is expected that preventive behaviors can be effectively promoted using policies and information sharing centering on the effective variables for each age group. Further, there was a significant effect from each of the variables related to affect, which means that there is the potential to increase the frequency of implementation of behaviors by changing various affect factors used in information sharing and adjusting the environment from a variety of perspectives.

Regarding vaccination, more than 60% of the respondents said, “I will wait and see,” with the reason being the emotional option of “fear of side effects” more often than “doubts about effectiveness.” Among the people who said, “I will get vaccinated as soon as possible,” the most common reason was “Because I do not want to keep living in fear of the risk of infection,” which indicates that people have different reasons for being positive or cautious about vaccination. These results suggest that the suitability of the information shared regarding vaccinations may vary according to the attitudes of the target group.

## 5. Conclusions

This study investigated whether risk perception and preventive behavior implementation regarding COVID-19 differ in response to different mortality rates depending on the age group and what factors determine preventive behaviors using questionnaire survey data. The analysis found that preventive behaviors were carried out according to the risk of COVID-19, and preventive behaviors were determined not only by risk perception but also by affect factors such as altruism and regret.

However, the results of this study have some limitations. First, since preventive behavior implementation was self-reported, it cannot be ruled out that social desirability may have an effect. Second, this survey collected limited information about personal demographics. Therefore, it could not examine the negative effects of low socioeconomic status, including low income, low education, and lack of insurance, on preventive behaviors [30].

The social situation and the actual risk of COVID-19 may change in the future with the onset of vaccination in Japan (as of May 2021), beginning with medical personnel and older people, and with the outbreak of variants and scientific discoveries reached on the basis of acquired data and so forth. It will be necessary to closely observe any changes in people’s attitudes and perceptions in line with these developments, including whether the behaviors of young people will change appropriately in case they are at a greater risk of aggravation. Moreover, in order to investigate policies and information sharing that promote preventive behaviors, it will be necessary to clarify the information and personal experiences and values that produce the attitudes and affect found to promote behavioral implementation in this study.

## Figures and Tables

**Table 1 ijerph-18-09979-t001:** Results of factor analysis of risk characteristics.

	Low Transmission Period		High Transmission Period	
	Factor 1	Factor 2	Factor 1	Factor 2
How great a threat to future generations will the damage be?	0.72	−0.22	0.78	−0.24
How easy is it to reduce risk?	0.67	−0.14	0.73	−0.20
How likely is it to lead to death?	0.62	0.11	0.66	0.05
To what extent do you have a scientific understanding of the risk?	0.58	−0.23	0.59	−0.06
Will one person die at a time from this, or will many people lose their lives at once?	0.53	0.06	0.63	0.01
Is the risk something new, or is it something that has been known for a long time?	0.06	0.61	0.04	0.63
How many people will be exposed to damage?	−0.17	0.55	−0.20	0.70

**Table 2 ijerph-18-09979-t002:** Results of factor analysis regarding anxiety.

	Low Transmission Period		High Transmission Period	
	Factor 1	Factor 2	Factor 1	Factor 2
Personally being infected with COVID-19	0.77	0.25	0.82	0.19
Pressure on medical institutions	0.76	0.23	0.82	0.15
Not knowing when the disease will be overcome	0.80	0.18	0.80	0.19
Time needed to develop and approve vaccines and medical treatments	0.72	0.30	0.68	0.30
Infecting others around me	0.70	0.26	0.74	0.26
Stress due to staying at home for long periods	0.16	0.72	0.15	0.74
Deterioration and diminishing of personal relationships due to a lack of communication with friends and colleagues	0.26	0.69	0.20	0.69
Changes in work and study style due to remote working, online lessons, etc.	0.20	0.62	0.14	0.66
Stress due to restrictions on eating out, traveling, entertainment, etc.	0.34	0.62	0.29	0.58
Deterioration of health due to lack of exercise	0.28	0.61	0.31	0.62
Education and exams for me or my children	0.12	0.61	0.09	0.56

**Table 3 ijerph-18-09979-t003:** Results of factor analysis of attitudes toward COVID-19.

	Low Transmission Period		High Transmission Period	
	Factor 1	Factor 2	Factor 1	Factor 2
COVID-19 exists in my immediate vicinity.	0.73	−0.05	0.75	−0.11
COVID-19 is a serious problem for me personally.	0.64	−0.14	0.72	−0.18
Regardless of one’s personal circumstances, in order to limit COVID-19, one should not leave home for nonessential and nonurgent reasons and should avoid the three Cs, etc.	0.53	0.02	0.64	−0.09
COVID-19 is an issue that transcends generations, regions, and countries.	0.79	−0.20	0.78	−0.21
Anyone can be infected no matter how many countermeasures they take.	0.61	0.02	0.60	0.00
I do not want to change my personal behavior to prevent COVID-19.	−0.11	0.63	−0.16	0.69
Individual behavior changes nothing in the control of COVID-19.	−0.19	0.71	−0.16	0.68
COVID-19 will be resolved using new technologies without individuals making major changes to their lifestyles.	0.04	0.52	−0.05	0.53
The rumors about the effect of COVID-19 on society and the economy are exaggerated.	−0.01	0.59	−0.03	0.52

**Table 4 ijerph-18-09979-t004:** Ratio of implementation of preventive behaviors.

	Low Transmission Period			High Transmission Period		
	18–39 y/o	40–59 y/o	60 y/o and above	18–39 y/o	40–59 y/o	60 y/o and above
Use of mask	86.8	94.3	95.4	89.8	93.5	96.2
Infection control measures, such as hand washing or cleaning fingers using an antiseptic solution	74.6	85.7	91.8	80.1	86.8	90.7
Use of gloves	11.4	7.5	10.4	14.0	13.3	22.0
Use of cashless payments	34.8	38.7	40.8	37.1	39.7	48.2
Refraining from leaving home for nonurgent or nonessential purposes	44.7	47.8	68.5	54.7	67.5	78.9
Postponing or canceling travel or leisure	33.6	43.4	58.1	44.2	57.1	70.8
Refraining from eating out	38.9	42.1	57.3	49.4	58.4	71.7
Having sufficient exercise, nourishment, and sleep	32.5	27.5	46.1	32.5	38.7	46.7
Refraining from experiential entertainment	24.0	29.9	33.4	26.9	35.8	42.3
Refraining from physical contact, including handshakes and hugs	26.0	34.3	46.7	24.9	43.1	51.6
Moving by car or bicycle rather than public transportation	21.4	24.9	39.1	18.4	31.4	38.1
Communication using online tools rather than face-to-face	12.9	10.9	8.9	13.4	16.4	16.9
Use of mail order and delivery services	10.5	10.9	16.3	17.8	18.2	18.4
Avoiding crowded locations and times to the extent possible	33.3	48.1	62.6	36.3	52.7	66.0
Nothing	9.7	4.2	2.1	5.6	4.4	1.9

**Table 5 ijerph-18-09979-t005:** Main factors that determine preventive behaviors.

	Estimate	T Value	
Risk characteristics: Critical factor	0.00	0.06	
Risk characteristics: Trivial factor	−0.21	−2.56	*
Anxiety: Direct anxiety factor	0.51	5.95	***
Anxiety: Indirect anxiety factor	0.21	3.38	***
Attitude: Cautious factor	0.61	7.77	***
Attitude: Skeptical factor	−0.58	−7.96	***
Reason effectiveness	−0.59	−7.92	***
Reason altruism	−0.45	−6.74	***
Reason risk perception	−0.44	−6.95	***
Reason regret	−0.62	−10.17	***
Reason conformity	−0.24	−3.74	***
Gender	−0.26	−4.71	***
Age	0.01	−4.71	**
Survey period	−0.39	−7.28	***
Partial	5.38	27.83	***
Adjusted R^2^	0.45		
*n*	2400		
Dependent variables: No. of implemented preventive behaviors	6.21		
(Reason) Dummy variable (Selected = 1, Not selected = 0) (Gender) Dummy variable (Male = 1, Female = 2) (Study period) Dummy variable (Low transmission period = 1, High transmission period = 2)			

* *p* < 0.05 ** *p* < 0.01 *** *p* < 0.001.

**Table 6 ijerph-18-09979-t006:** Main factors that determine preventive behaviors (by survey period).

	18–39 y/o			40–59 y/o			60 y/o and above		
	Estimate	T Value		Estimate	T Value		Estimate	T Value	
Risk characteristics: Critical factor	0.15	1.08		−0.18	−1.34		0.06	0.50	
Risk characteristics: Trivial factor	−0.10	−0.62		−0.36	−2.69	**	−0.09	−0.70	
Anxiety: Direct anxiety factor	0.45	2.95	**	0.82	5.42	***	0.26	1.77	
Anxiety: Indirect anxiety factor	0.20	1.68		0.11	0.99		0.31	3.03	**
Attitude: Cautious factor	0.77	5.40	***	0.45	3.38	***	0.63	4.58	***
Attitude: Skeptical factor	−0.67	−4.82	***	−0.28	−2.12	*	−0.79	−6.75	***
Reason effectiveness	−0.62	−4.77	***	−0.50	−3.99	***	−0.63	−4.68	***
Reason altruism	−0.40	−3.34	***	−0.45	−3.83	***	−0.42	−3.68	***
Reason risk perception	−0.43	−3.07	**	−0.37	−3.14	**	−0.52	−5.79	***
Reason regret	−0.73	−5.91	***	−0.72	−6.47	***	−.52	−5.65	***
Reason conformity	−0.32	−2.67	**	−0.08	−.67		−0.30	−2.91	**
Gender	−0.21	−1.98	*	−0.32	−3.32	**	−0.26	−3.05	**
Survey period	−0.22	−2.13	*	−0.56	−5.97	***	−0.38	−4.52	***
Partial	5.90	35.43	***	5.96	40.04	***	6.33	42.09	***
Adjusted R^2^	0.45			0.44			0.36		
*n*	684			770			946		
Dependent variables: No. of implemented preventive behaviors	5.14			5.99			7.17		

* *p* < 0.05 ** *p* < 0.01 *** *p* < 0.001.

## Data Availability

The data presented in this study are available on request from the corresponding author.

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
