# Peer review of "Determinants of Preventive Behaviors for COVID-19 in Japan"

_ijerph, 2021, doi:10.3390/ijerph18199979_

Round 1

Reviewer 1 Report

Thank you for developing a well-rounded research question. I have only two main major comments that are methodology-related and two introduction points.

Introduction

  1. I believe it's important to discuss risk management behavior that is already in Japan that might differ from other countries. For example, willingness to wear masks.
  2. The research question is not very clear on what you are attempting to examine. I recommend rewording it to make it clear what the research question is.

Methodology

  1. Nationally representative sample: what was the methodology? I am having a hard time understanding how this is a nationally representative sample if sample weights are not applied?
  2. The questionnaire consisted of multiple sales. Were any of them validated? If not, then please provided as a limitation of this study.

Major strengths of the paper

  1. two time periods with high and low transmission
  2. An introduction that describes the importance of the study
  3. factor analysis of the scales.
  4. Discussion grounded in the findings.

Author Response

Thank you very much for reviewing my paper and comments.
According to your comments, I have responded as follows.

Introduction:

1)The difference between Japan and other countries in terms of risk management behavior is that nothing is mandatory in Japan. All risk management actions are required and there are no penalties for breaking them. I added about this point in the preamble as a characteristic of Japan: L27-29.

However, unlike other countries, Japanese government asks the public to wear masks and refrain from going out, but this is just request, not obligatory and there are no penalties.

2) As you pointed out, the purpose of this study was not clear, so I corrected it as follows (L78-83).

The purpose of this study is to identify factors that may contribute to the promotion of preventive behavior in COVID-19. Specifically, this study will investigate whether people's risk perceptions corresponded to actual risk assessments of age groups, and whether risk management behaviors such as hand washing and wearing masks depended on actual risk assessments. Moreover, it reveals not only risk perceptions and attitudes, but also the role of emotional impact on the implementation of risk management behaviors.

Methodology:

1)This survey used a quota sampling instead of weight back method for representativeness. Corrected L94-95 and clarified the sampling method.

Respondents were collected using quota sampling based on demographics by prefecture, gender, and age.

2)I'm so sorry, but I didn't understand your comment.

As you mentioned, the survey uses the SD method, the 5-case method (Likert scale), and multiple selection.

I would appreciate it if you could explain a little more about how to perform a "validation" on these questions.

Reviewer 2 Report

Authors did a fine job of presenting their findings in a very scientific method; study design was adequately communicated; the results section is rather overwhelming with extensive data acquisition--authors should attempt to succinctly present the data, perhaps using graphs rather than tables as reported in this manuscript. There are some English grammar deficiency throughout the prose; please review these areas and make corrections in sentence structure (line 13 --> ..."any they more..."), (line 26 --> ..."including avoiding..."), (line 70 --> DELETE "to do this").  Lines 71-75 appears tone the study "aim" however it is disorganize and not as clear as it should be to represent the underlying "aim"of this study.  Throughout the results section should present findings only and not interpretations (these should be included in the discussion NOT results; the results show the data whereas discussion prevents interpretation of the data).

Overall, a nicely done work on a very relevant topic; publication of this manuscript can serve to be informative to the global scientific and academic community.

Author Response

Thank you very much for reviewing my paper and comment.
According to your comments, I have responded as follows.

As you pointed out, this paper has many tables, and I thought that it should be made into a figure, but only Table 4 can be replaced with a figure. Since there are 6 data of 3 age groups in 2 surveys, I thought that it might be more difficult to understand than the table in the figure, so I left it as the table.

English grammar:

Line 13: I corrected.

Line 26: I corrected.

Line 70: Since I revised aim of this study, the commented part was disappeared. I corrected purpose as follows (L78-83).

The purpose of this study is to identify factors that may contribute to the promotion of preventive behavior in COVID-19. Specifically, this study will investigate whether people's risk perceptions corresponded to actual risk assessments of age groups, and whether risk management behaviors such as hand washing and wearing masks depended on actual risk assessments. Moreover, it reveals not only risk perceptions and attitudes, but also the role of emotional impact on the implementation of risk management behaviors.

The interpretation contained in the results report removed: L195-198, l263-265, l272-274. The interpretation given there was already included in the discussion, so I have not added it to the discussion.

Please see the attachment for revised version. 
